Successive DNA extractions improve characterization of soil microbial communities

Dimitrov Mauricio R. 1 2 M.Dimitrov@nioo.knaw.nl
http://orcid.org/0000-0001-6286-7484 Veraart Annelies J. 1
de Hollander Mattias 1
Smidt Hauke 2
van Veen Johannes A. 1
http://orcid.org/0000-0001-6701-8668 Kuramae Eiko E. 1
1 Department of Microbial Ecology, Netherlands Institute of Ecology (NIOO-KNAW) , Wageningen , Netherlands
2 Laboratory of Microbiology, Wageningen University , Wageningen , Netherlands
Landa Blanca B.
Electronic publication date: 2017 Feb 1
Publication date: 2017
Volume: 5
Electronic Location ID: e2915
Received 2016 Jun 27; Accepted 2016 Dec 15
Copyright: © 2017 Dimitrov et al.
Copyright year: 2017
Copyright holder: Dimitrov et al.
License: This is an open access article distributed under the terms of the Creative Commons Attribution License, which permits unrestricted use, distribution, reproduction and adaptation in any medium and for any purpose provided that it is properly attributed. For attribution, the original author(s), title, publication source (PeerJ) and either DOI or URL of the article must be cited.
License URL: https://creativecommons.org/licenses/by/4.0/

Keywords: Soil DNA extractions, Soil microbial characterization, 454-pyrosequencing, T-RFLP, Soil microbial abundance, Soil bacterial community, Soil fungal community

Funding: Strategic Research Fund of the WIMEK graduate school Netherlands Organisation for Scientific Research (NWO) 823.001.008 Mauricio R. Dimitrov was supported through funding from the Strategic Research Fund of the WIMEK graduate school (project “Adaptive capacity and functionality of multi-trophic aquatic ecosystems”). Annelies J. Veraart was supported by a grant (823.001.008) of the Netherlands Organisation for Scientific Research (NWO). The funders had no role in study design, data collection and analysis, decision to publish, or preparation of the manuscript.

==============================
Currently, characterization of soil microbial communities relies heavily on the use of molecular approaches. Independently of the approach used, soil DNA extraction is a crucial step, and success of downstream procedures will depend on how well DNA extraction was performed. Often, studies describing and comparing soil microbial communities are based on a single DNA extraction, which may not lead to a representative recovery of DNA from all organisms present in the soil. The use of successive DNA extractions might improve soil microbial characterization, but the benefit of this approach has only been limitedly studied. To determine whether successive DNA extractions of the same soil sample would lead to different observations in terms of microbial abundance and community composition, we performed three successive extractions, with two widely used commercial kits, on a range of clay and sandy soils. Successive extractions increased DNA yield considerably (1–374%), as well as total bacterial and fungal abundances in most of the soil samples. Analysis of the 16S and 18S ribosomal RNA genes using 454-pyrosequencing, revealed that microbial community composition (taxonomic groups) observed in the successive DNA extractions were similar. However, successive DNA extractions did reveal several additional microbial groups. For some soil samples, shifts in microbial community composition were observed, mainly due to shifts in relative abundance of a number of microbial groups. Our results highlight that performing successive DNA extractions optimize DNA yield, and can lead to a better picture of overall community composition.

Introduction

Microorganisms are key to various biogeochemical processes that drive life on Earth (Falkowski, Fenchel & Delong, 2008). Soil is one of the most diverse biomes found on Earth and a large reservoir of microbial diversity (Bardgett & van der Putten, 2014; Gans, Wolinsky & Dunbar, 2005; Torsvik, Goksøyr & Daae, 1990; Torsvik, Øvreås & Thingstad, 2002). Besides being essential drivers of biogeochemical processes, soil microorganisms also play an important role in processes such as plant nutrition (Mendes, Garbeva & Raaijmakers, 2013), disease suppression (Mendes et al., 2011), bioremediation (Maphosa et al., 2012), global warming mitigation (Bender et al., 2014), to just name a few. However, understanding the mechanisms behind all these processes is not an easy task, since the vast majority of microorganisms are still unculturable (Hawksworth & Rossman, 1997; Torsvik & Øvreås, 2002). The introduction of culture independent methodologies has revolutionized the way soil microbial communities are studied. Extracting and characterizing DNA has become ordinary in most soil microbial ecology studies (Delmont et al., 2012; Navarrete et al., 2015; Pan et al., 2014; Tahir et al., 2015). Moreover, constant improvements and accessibility of high throughput sequencing technologies have allowed researchers to characterize soil microbial communities in an unprecedented way and at ecologically relevant scales and resolution of time, space and environmental conditions.

Due to its stability, DNA is often the nucleic acid of choice to be used to characterize microbial communities in soils. Once extracted, DNA can be used in a range of experiments that may provide insights with respect to the abundance, diversity and functional potential of soil microbial communities. Therefore, successful characterization of soil microbial communities is directly dependent on the quality of the DNA extracted from such soil sample. With the introduction of culture independent methodologies to study soil microbial communities, a variety of soil DNA extraction protocols have been developed (Berry et al., 2003; Bürgmann et al., 2001; Liles et al., 2008; Robe et al., 2003; Zhou, Bruns & Tiedje, 1996). However, DNA extraction from soil can be laborious and problematic (Braid, Daniels & Kitts, 2003; Dong et al., 2006; Frostegård et al., 1999; Robe et al., 2003). Often, as an alternative to simplify and standardize procedures, commercial DNA extraction kits are used. Comparison of different soil DNA extraction protocols, including commercial kits, has shown that DNA yield and purity varies greatly depending on the protocol and soil type (Bürgmann et al., 2001; Inceoglu et al., 2010; Knauth, Schmidt & Tippkötter, 2013). Therefore, every DNA extraction protocol has its own bias, and it will yield DNA that is representative of a portion of the microbial community present in the original soil sample (Delmont et al., 2011). Feinstein, Sul & Blackwood (2009) analyzed DNA extraction efficiency of a commonly used commercial soil DNA extraction kit, and indicated that not all microbial DNA present in a soil sample is extracted with a single DNA extraction. When soil DNA, from successive DNA extractions performed on a single sample, was used to characterize microbial communities, substantial shifts in the bacterial community were observed. Although the findings of Feinstein, Sul & Blackwood (2009) were published more than seven years ago, less than 15% of recently published studies have used multiple DNA extractions when characterizing soil microbial communities (Fig. S1). Here, to investigate further how bias of incomplete soil DNA extraction may affect microbial characterization, we expanded on the reported findings by Feinstein, Sul & Blackwood (2009). Two widely used commercial soil DNA extraction kits were used to extract DNA from a variety of soils. Successive DNA extractions were performed on six different soils collected throughout the Netherlands, and bacterial and fungal abundances, as well as community diversity and composition of each successive extraction, were assessed by using next generation technology sequencing of 16S and 18S ribosomal RNA (rRNA) gene amplicons.

Materials and Methods

Soil samples

Soil samples as well as sampling procedure used in the present work have been described previously (Kuramae et al., 2012). Soil cores (8 cm of diameter and 30 cm deep) were sampled in six contrasting fields located in different regions throughout the Netherlands (Fig. S2). Briefly, a central sampling point and four other points, located 20 m from the central point, were chosen in each field, which resulted in five sample points per field (i.e., A, B, C, D and E). Each individual sample was composed of five subsamples (i.e., A1, A2, A3, A4, A5; B1, B2, B3, etc.), which were taken randomly within a two-meter radius in each sample point (A, B, C, D and E). Equal amounts of the five subsamples were pooled to obtain a single sample per sampling point (A, B, C, D and E), resulting in five biological replicates per field. Soil samples were chosen to represent five of the most important land management practices in the Netherlands (conventional and organic arable field, pasture, pine and deciduous forest). Moreover, soil samples were chosen and separated according to sand and clay content, resulting in two different soil classes: ‘sandy’ and ‘clay’ (Table S1). After sampling, samples were stored at −80 °C until further processing.

DNA extraction and quantification

Soil DNA was extracted from each soil sample using two different commercial kits, which are widely used for such purpose (Kuramae et al., 2012; Martin-Laurent et al., 2001; Mendes et al., 2014; Sutton et al., 2013). Three replicates were extracted from each soil sample. The PowerSoil DNA isolation kit (MoBio Laboratories, Carlsbad, CA, USA) and the FastDNA Spin kit for soil (FS) (MP Biomedicals, Solon, OH, USA) were used according manufacturer’s instructions. The bead-beating step of the PowerSoil DNA isolation kit (PS) was done at 5.5 m s−1 for 10 min, using a Retsch MM301 mixer mill (Retsch GmbH, Haan, Germany). Samples extracted with the FastDNA Spin kit for soil (FS) were processed using a FastPrep24 instrument (MP Biomedicals, Solon, OH, USA). For both kits, an initial DNA extraction was followed by a successive extraction, which was then followed by another extraction after samples had been stored overnight at −20 °C (Fig. 1). Therefore, three DNA extractions were performed on three replicates of the six soil samples, resulting in nine DNA extracts per soil and extraction kit. After the bead-beating step of the first extraction (E1), tubes containing beads and soil were kept on ice until extraction had been finished. To start the second extraction (E2), while using PS, solution from new PowerBead tubes, without beads, was added to PowerBead tubes used in the first extraction, which still contained soil and beads. After that, DNA extraction proceeded exactly in the same way as for E1. After the bead-beating step of E2, tubes containing beads and soil were store overnight at −20 °C. The third and final extraction (E3) was performed as described for E1 and E2. The procedure described for PS was also used for FS; when initiating a new DNA extraction sodium phosphate buffer and MT buffer were added to the lysis matrix tubes containing beads and soil. Volumes of added buffers were always in line with manufacturer’s instructions. Extraction proceeded normally afterwards. Supernatant recovery, throughout the whole DNA extraction procedure, was done carefully in order to obtain a complete recovery and minimize carryover of DNA from one extraction to another. When higher volumes of supernatant were recovered, compared to manufacturer’s instructions, adjusted volumes of solutions were used in order to maintain the proper concentration of reagents. Total DNA quantity and quality were measured using a Qubit 2.0 fluorometer (Life Technologies, Carlsbad, CA, USA) and a NanoDrop 1000 spectrophotometer (Thermo Scientific, Wilmington, DE, USA), as well as visualized on 1% (w/v) agarose gel under UV light after staining with ethidium bromide. Tris-acetate-EDTA (TAE) buffer was used for gel preparation and electrophoresis.

Figure 1 Successive DNA extraction procedure.

Scheme represents extraction performed with the PowerSoil DNA isolation kit (PS). The same procedure was used for FastDNA Spin kit for soil (FS), however, the amount of soil and bead-beating step were different.

Bacterial and fungal abundances

Quantitative real time PCR was used to determine total bacterial and fungal abundances in each soil sample by targeting the 16S and 18S rRNA genes, respectively. Quantitative PCR (qPCR) reactions were performed in a 384-well plate (Bio-Rad, Hercules, CA, USA) using a CFX384 Real-Time PCR Detection system (Bio-Rad). All samples were analyzed in triplicate, and reactions were carried out in a total volume of 10 μL. qPCR reactions were prepared using 5 μL of iQ SYBR Green super mix (Bio-Rad), 0.4 μL of forward and reverse primers (10 μM), 0.1 μL of bovine serum albumin (20 mg/mL), 0.1 μL of VisiBlueTM qPCR mix colorant (TATAA Biocenter, Gothenburg, Sweden) and 4 μL of DNA (2.5 ng/μL). Primer combinations and cycle conditions are described in Table 1. At the end of each qPCR run, a melting curve analysis was performed from 60 to 99 °C with an increase of 0.5 °C every 10 s. Purity of the qPCR products was checked by the observation of a single peak on the melting curve, while correct size of the amplicons was confirmed on a 1% (w/v) agarose gel. For each qPCR reaction, a standard curve comprising serial 10-fold dilutions of the target gene was created. Standards were obtained by amplifying the target genes from the following sources: Escherichia coli (16S rRNA gene) and Aspergillus niger (18S rRNA gene).

Table 1 Adaptors and primers used for targeting prokaryotic and fungal community.

Primers	Sequence 5′–3′	Target	Application	Cycle conditions	References	
ITS1F	TCCGTAGGTGAACCTGCGG	Fungi	T-RFLP	95 °C—5 min; 35 cycles of 95 °C—30 s, 55 °C—40 s, 72 °C—90 s	White et al. (1990)	
ITS4R	TCCTCCGCTTATTGATATGC	Fungi	T-RFLP	White et al. (1990)	
BACT1369F	CGGTGAATACGTTCYCGG	Bacteria	qPCR	95 °C—3 min; 40 cycles of 95 °C—30 s, 56 °C—45 s, 72 °C 60 s.	Suzuki, Taylor & DeLong (2000)	
PROK1492R	GGWTACCTTGTTACGACTT	Bacteria	qPCR	Suzuki, Taylor & DeLong (2000)	
FF390	CGWTAACGAACGAGACCT	Fungi	qPCR	95 °C—3 min; 40 cycles of 95 °C—30 s, 52 °C—45 s, 72 °C—60 s	Modified from Vainio & Hantula (2000)	
FFR1	AICCATTCAATCGGTAIT	Fungi	qPCR	Vainio & Hantula (2000)	
515F	GTGCCAGCMGCCGCGGTAA	Archaea and bacteria	Sequencing	95 °C—2 min; 25 cycles of 95 °C—30 s, 53 °C—45 s, 72 °C 60 s.	Bates et al. (2011)	
806R	GGACTACVSGGGTATCTAAT	Archaea and bacteria	Sequencing	Bates et al. (2011)	
FF390	CGWTAACGAACGAGACCT	Fungi	Sequencing	95 °C—2 min; 25 cycles of 95 °C—30 s, 52 °C—45 s, 72 °C—60 s	Modified from Vainio & Hantula (2000)	
FR1	AICCATTCAATCGGTAIT	Fungi	Sequencing	Vainio & Hantula (2000)	

Fungal community analysis by T-RFLP

Terminal restriction fragment length polymorphism (T-RFLP) was used to determine whether the fungal community of a soil sample differed among the three successive DNA extractions performed. Internal transcribed spacer (ITS) regions were used to target fugal community by amplifying ITS1 region, 5.8S rRNA gene and ITS2 region using specific primers (Table 1). A single replicate from each soil sample was randomly chosen to determine the fungal community by T-RFLP. Three PCR reactions were performed for each sample. PCR was performed with final reaction volume of 25 μL, which contained 2.5 μL of 10× PCR reaction buffer with 20 mM of MgCl2 (Roche Applied Sciences, Indianapolis, IN, USA), 200 μM of dNTPs, 1 μM of each primer, 1.25 U of Fast Start DNA polymerase (Roche Applied Sciences, Indianapolis, IN, USA) and 5 μL of template DNA (5 ng/μL). Cycle conditions are given in Table 1. The forward primer was labelled with the fluorescent dye 6-FAM, while the reverse primer was labelled with NED (Applied Biosystems, Foster City, CA, USA). Successful amplification was confirmed by running PCR products on a 1.5% (w/v) agarose gel. PCR products were digested with 10 units of HhaI (Thermo Scientific, Wilmington, DE, USA) at 37 °C for 3 h. Enzyme inactivation was performed by incubation at 80 °C for 20 min. After inactivation, digested PCR products were purified using ethanol precipitation. Appropriate dilutions, based on test runs of terminal restriction fragments (TRFs), were analyzed with an ABI 3130 sequencer using GeneScan™—500 LIZ (Applied Biosystems, Foster City, CA, USA) as a size standard.

Prokaryotic and fungal community analyses by 454-pyrosequencing

Prokaryotic and fungal community composition, as well as diversity were investigated by 454-pyrosequencing (454 Life Sciences, Roche). The prokaryotic community (bacterial and archaeal communities) was targeted by amplification of V3 and V4 regions of the 16S rRNA gene, while the fungal community was assessed by amplification of V7 and V8 regions of the 18S rRNA gene. Amplicons were generated by PCR reactions of 50 μL (total volume) containing 5 μL of 10× PCR reaction buffer with 20 mM of MgCl2 (Roche Applied Sciences, Indianapolis, IN, USA), 200 μM of dNTPs, 1 μM of each primer, 1.25 U of Fast Start DNA polymerase (Roche Applied Sciences, Indianapolis, IN, USA) and 5 μL of template DNA (5 ng/μL). PCR reactions were performed in duplicate, which were mixed prior to purification. As for T-RFLP, only a single replicate of each sample was used (same replicate as used for T-RFLP). Primer sequences and cycle conditions are shown in Table 1. Confirmation of amplification was performed by electrophoresis of PCR products on a 1.5% (w/v) agarose gel. PCR products were purified using the GeneJET PCR Purification Kit (Thermo Scientific, Wilmington, DE, USA), and quantified using a Qubit 2.0 Fluorometer (Life Technologies). Purified and quantified PCR products were then mixed in equimolar amounts at a final concentration of 500 ng/μL. The equimolar mixture was purified by electrophoresis of pooled amplicons in a 1.5% (w/v) agarose gel and excision of the band. DNA was recovered from agarose using the GeneJET Gel Extraction Kit (Thermo Scientific, Wilmington, DE, USA). Gel purified amplicons were sequenced using an FLX genome sequencer in combination with titanium chemistry (Macrogen Inc., Seoul, South Korea).

Data analysis

Differences in DNA yield, bacterial and fungal abundances, represented as percentage of recovery after first, second and third extraction, were analyzed by two-way ANOVA using IBM SPSS Statistics for Macintosh software, version 23 (IBM Corp., Armonk, NY, USA). Similarly, a two-way ANOVA was used to analyze differences in total DNA yield as well as bacterial and fungal abundances observed after all three successive extractions. T-RFLP analyses were performed according to Mendes et al. (2012). Briefly, profiles were analyzed using PeakScanner v1.0 software (Applied Biosystems, Foster City, CA, USA), and TRFs of less than 50 bp and bigger than 800 bp were excluded. The relative abundance of a single TRF was calculated as percent fluorescence intensity relative to total fluorescence intensity of the peaks (Culman et al., 2008). An average of three replicates was calculated for each individual sample. Principal component analysis (PCA) was used to assess differences in fungal community composition between successive DNA extractions of a single sample, within a single DNA extraction kit. Multivariate statistical analysis was performed using Canoco 4.5 software (ter Braak & Šmilauer, 2002).

Sequencing analyses were performed using a Snakemake workflow (Koster & Rahmann, 2012), which follows a standard operating procedure for 454 data in mothur version 1.33.2 (Schloss et al., 2009). Flowgrams were demultiplexed allowing two mismatches on the barcodes and three mismatches on the forward primer, flowgrams were trimmed to a size of 635 flows. Flowgrams were corrected using the shhh.flows command, which is a mothur implementation of the original PyroNoise algorithm (Quince et al., 2011). Afterwards, results of the different sff files were combined for further analysis. Merged sequences were aligned and classified with SINA (Pruesse, Peplies & Glockner, 2012) against the SILVA 115 database (Quast et al., 2013). After alignment, some reads did not align to the same region as most of the reads. Therefore, reads were kept if containing at least 90% of its sequence aligned to a region common to all reads. To reduce sequence errors, sequences that were within two mismatches of each other were merged. Chimeric sequences were identified and removed using UCHIME (Edgar et al., 2011). Operational taxonomic units (OTUs) were formed at maximum distance of 0.03 using the dist.seqs command and average neighbor clustering. For each OTU a consensus taxonomy was determined using the classify.otu command. Representative sequences for each OTU were re-aligned to the SILVA reference alignment, and a neighbor joining tree was created using the clearcut program (Sheneman, Evans & Foster, 2006). Taxonomic classification and OTU clustering data were combined into the BIOM format (McDonald et al., 2012) for further downstream analyses with the Phyloseq (McMurdie & Holmes, 2013) package for R (R Development Core Team, 2014). Due to a low quality sequencing output, E2 performed with FS on sample sandy 3 was removed from the fungal dataset after sequence quality control. Numbers of reads were not rarefied among samples (normalized to the lowest number of reads) before clustering analysis, since recent work has shown that rarefying is unnecessary (McMurdie & Holmes, 2014). Instead, OTU raw abundances were transformed to relative abundances prior analysis, which has been shown to be an alternative to rarefying (McMurdie & Holmes, 2014). Prior to clustering analysis, sample-wise singletons and doubletons were discarded, where an OTU would be kept only if observed in at least one sample and contained at least three reads.

Diversity analyses were performed using OTU tables that had been rarefied to the lowest number of reads and included singletons and doubletons. The lowest number of reads for the 16S rRNA dataset was 3,180, whereas for the 18S rRNA dataset it was 3,440. An additional filtering step was done only for the 18S rRNA gene dataset, where only reads belonging to the kingdom Fungi were kept for downstream analyses. Raw sequencing data obtained from 454-pyrosequencing were submitted to the NCBI sequence read archive (SRA) under the accession numbers SRR5040745 (16S rRNA reads) and SRR5043664 (18S rRNA reads). For all statistical tests performed, statistical significance was accepted at p < 0.05.

Results

DNA yield

Overall, DNA extraction with both commercial kits yielded a substantial amount of DNA, regardless of soil type. Efficiency was higher when using the FS than the PS DNA extraction kit, independent of soil type. FS yielded around four times more DNA from clay soils and around three times more DNA from sandy soils than PS (Fig. 2; Table S2). For almost all soil samples, DNA extractions performed with both kits yielded the highest DNA concentration in E1, except for clay 2 (PS extraction) and sandy 3 (extraction performed with both kits), which showed the highest DNA yield in E2 (Table S2). DNA extraction of clay 1 was similar for both kits, where the highest amount of DNA was extracted in E1 (around 60% of total DNA obtained), with E2 and E3 showing lower DNA concentration compared to previous extractions (Figs. 2A and 2C; Table S2A). Less than half of the total DNA obtained from clay 2 was extracted in E1 for both kits, with E2 and E3 still yielding substantial amounts of DNA (Figs. 2A and 2C; Table S2A). Clay 3 showed distinct DNA extraction patterns when comparing both kits (Figs. 2A and 2C). Using PS, around 95% of total DNA extracted was recovered in E1 (Fig. 2A). However, when using FS, only 40% of the total DNA extracted was recovered in E1 (Fig. 2C). Among clay soils, regardless of the kit used, clay 3 yielded the highest total DNA concentration followed by clay 1 and clay 2. The same trend was also observed for sandy soils, where sandy 1, a pasture soil as clay 3, yielded the highest total DNA concentration, followed by sandy 3 and sandy 2 (Figs. 2B and 2D; Table S2B). DNA extractions of sandy 1 showed a similar pattern regardless the kit used, with 55–60% of DNA obtained in E1 (Figs. 2B and 2D; Table S2B). The same was observed for sandy 2, with approximately 55% of the total DNA recovered being extracted in E1 (Figs. 2B and 2D; Table S2B). Comparing all three successive extractions, sandy 3 showed the highest DNA concentration in E2, for both kits (Figs. 2B and 2D; Table S2B). Approximately 50% of the total DNA extracted was obtained in E2, with E1 yielding more DNA than E3. Different soil management influenced the total amount of DNA obtained, regardless of DNA extraction methodology. Pasture soil yielded a considerably higher amount of total DNA compared to other managements, with pine forest soil showing the least amount of DNA. DNA extraction kit had a significant effect on the total DNA yield, with FS extracting significantly more DNA than PS (p < 0.05), however, soil type did not significantly influence DNA yield (p > 0.05). No significant interaction was observed between DNA extraction kit and soil type (p > 0.05). Neither soil type nor DNA extraction kit had a significant effect on DNA yield in E1, E2 and E3 (recovery percentage) (p > 0.05).

Figure 2 Cumulative DNA yields in successive extractions of clay (C1–C3) and sandy soils (S1–S3) using PS and FS DNA extraction kits.

Average and error bars (SD) of all three biological replicates are presented for each DNA extraction. (A) Clay soils extracted with PS; (B) sandy soils extracted with PS; (C) clay soils extracted with FS; (D) sandy soils extracted with FS. DNA quantification was performed using a Qubit 2.0 fluorometer.

Bacterial and fungal abundances in successive DNA extractions

Overall, qPCR results of both targeted genes (16S and 18S rRNA) indicated similar patterns as observed in DNA quantification of all soil samples (Figs. 2–4). Bacterial and fungal abundances in clay 1 decreased with successive extractions, for both DNA kits, where no substantial increase in cumulative abundance could be seen after E2 (Figs. 3A, 3C, 4A and 4C). Interestingly, clay 2 presented almost constant bacterial and fungal abundances in all three DNA extractions, which resulted in a linear cumulative abundance increase with successive extractions for both kits (Figs. 3A, 3C, 4A and 4C). Differently from the other clay soils, clay 3 showed contrasting results when comparing DNA extraction kits. When using PS, bacterial and fungal abundances in E2 and E3 were very low compared to E1, indicating that almost all bacterial and fungal DNA available had been already extracted in E1 (Figs. 3A and 4A). However, when using FS, bacterial and fungal abundances as measured in E2 and E3 were considerably higher compared to E1 (Figs. 3C and 4C), indicating that substantial amounts of bacterial and fungal DNA were still present in the soil sample after E1. As observed for total DNA yield, bacterial and fungal abundances in clay soils were higher in samples extracted with FS compared to PS. This was not observed for sandy soils, where bacterial and fungal abundances were always higher in soil samples that had DNA extracted with PS. Irrespective of the kit used, bacterial and fungal abundances in sandy 1 samples were highest in E1 (Figs. 3B, 3D, 4B and 4D). However, when using PS, around only half of total bacterial and fungal DNA was recovered in E1, whereas, when using FS, these values were around 80%. A very distinct pattern was observed for bacterial and fungal abundances in sandy 2, when comparing DNA extraction kits. Samples extracted with PS had a recovery of around 45% of total bacterial and fungal DNA in E1, whereas samples extracted with PS showed a recovery of more than 80% of total bacteria and fungi abundance in E3 (Figs. 3B, 3D, 4B and 4D). After all three successive extractions, cumulative bacterial and fungal abundances were five times higher in samples extracted with PS compared to those extracted with FS, even with DNA yield being higher in samples extracted with FS. The total number of bacterial 16S rRNA gene copies in sandy 3 was similar for both kits (Figs. 3B and 3D). The same was observed for total fungal abundance. Total bacterial and fungal abundances were slightly higher in samples extracted with PS, however, samples extracted with either kit presented higher fungal abundance in E2 and E3 compared to E1 (Figs. 4B and 4D), which was not noticed for bacterial abundance. As observed for sandy 1 and sandy 2, a higher total DNA yield in samples of sandy 3, which had been extracted with FS, did not result in higher bacterial and fungal abundances. Such contrasting result was particular of sandy soils, since clay soils with higher DNA yield presented higher bacterial and fungal abundances. Different DNA extractions kits did not affect significantly bacterial nor fungal abundance in clay and sandy soils (p > 0.05). However, bacterial abundance was significantly affected by soil type (p < 0.05).

Figure 3 Cumulative 16S rRNA copy numbers in successive extractions of clay (C1–C3) and sandy soils (S1–S3) using PS and FS DNA extraction kits.

Average and error bars (SD) of all three technical replicates are presented for each DNA extraction. (A) Clay soils extracted with PS; (B) sandy soils extracted with PS; (C) clay soils extracted with FS; (D) sandy soils extracted with FS.

Figure 4 Cumulative 18S rRNA copy numbers in successive extractions of clay (C1–C3) and sandy soils (S1–S3) using PS and FS DNA extraction kits.

Average and error bars (SD) of all three technical replicates are presented for each DNA extraction. (A) Clay soils extracted with PS; (B) sandy soils extracted with PS; (C) clay soils extracted with FS; (D) sandy soils extracted with FS.

The use of different commercial kits for DNA extraction did neither affect bacterial nor fungal abundance (percentage of recovery) observed after first, second and third extractions (p > 0.05). However, Soil type had a significant effect on the total bacterial abundance (p < 0.05) and fungal abundance (p < 0.05) only after the first and third extraction.

Fungal community analysis by T-RFLP

Fungal community composition was initially measured by T-RFLP analysis, and PCA analysis was used to assess differences among different extractions of a single sample, within a single DNA extraction kit and soil type. Since forward and reverse primers were labelled, both were analyzed to investigate whether results were consistent independently of primer use. Analyses of both primers showed similar results for both DNA extraction kits and both soil types (Figs. S3 and S4). Fungal communities detected in clay soils showed to be distinct among different land management types, regardless of the DNA extraction kit used (Fig. S3). Overall, successive DNA extractions yielded different fungal communities in all clay soils. With the use of PS, clay 1 and clay 2 showed clearly different fungal communities when comparing the three extractions. However, for clay 3 fungal community composition present in E1 was different from E2 and E3, which were similar to each other (Figs. S3A and S3B). Results obtained with FS were slightly different from PS. Clay 1 presented a different fungal community in each successive DNA extraction. However, clay 2 showed a similar fungal community present in E1 and E2, which was different from the community observed in E3. Fungal communities detected for clay 3 were similar for all three extractions (Figs. S3C and S3D). As observed for clay soils, fungal communities observed in sandy soils were also distinct among different land management, independently of the DNA extraction kit used (Fig. S4). However, differently from clay soils, successive DNA extractions of sandy soils yielded similar fungal communities, especially when using PS (Figs. S4A and S4B). Total DNA extraction with PS revealed a very different fungal community when comparing different sandy soils, but not within soils. Successive DNA extraction with FS revealed a different fungal community only for sandy soils 1 and 2, with sandy 1 showing a similar community on the first two extractions, whereas sandy 2 presented a different community in all three extractions (Figs. S4C and S4D).

Prokaryotic and fungal community analysis by 454-pyrosequencing

To investigate further whether successive DNA extractions of the same soil sample yield different prokaryotic and fungal communities, as well as to confirm T-RFLP results (fungal community), next generation sequencing was performed on all three extractions obtained from each soil. The Chao1 index was used to measure species richness, whereas community diversity was described by Shannon index. Overall, apparent prokaryotic species richness in clay soils increased in E2 or E3 (Fig. 5A), with clay 1 and clay 3 (PS) being the only two samples that showed a decrease in species richness with successive extractions. Observed prokaryotic community diversity decreased with successive DNA extractions (Fig. 5A), regardless of the DNA extraction kit used. However, for clay 1 (PS) and clay 2 (FS) diversity increased in E3, compared to the previous extraction. Observed fungal species richness varied considerably across clay soils and DNA extraction kit (Fig. 6A). From all samples extracted with FS, clay 2 was the only sample that did not have its highest species richness value in E1. The same trend was observed for samples extracted with PS, where the highest species richness value was observed in E1. Observed fungal community diversity either increased or remained the same with successive extractions (Fig. 6A), regardless of the extraction kit used. Clay 1 (FS) was the only sample, within clay soils, that presented a decrease in fungal diversity with successive DNA extractions (Fig. 6A). Overall, observed prokaryotic species richness in sandy soils decreased with successive extractions, with sandy 1 (PS and FS) being the only sandy soil where species richness was highest not in E1 (Fig. 5B). Similar to clay soils, observed prokaryotic diversity in sandy soils decreased with successive extractions (Fig. 5B). Sandy 3 (PS) was the only exception, where the highest diversity was observed in E2. The highest observed fungal species richness value in most of the sandy soil samples was obtained only in E2 or E3 (Fig. 6B). Sandy 1 (FS) and sandy 2 (PS) were exceptions and had their highest species richness value in E1. Observed fungal diversity in sandy soils was similar to clay soils, where diversity increased with successive DNA extractions (Fig. 6B). The only exception was sandy 1 (FS), which showed a decrease in observed diversity with successive extractions (Fig. 6B).

Figure 5 Species richness (Chao1) and diversity (Shannon) of the prokaryotic community in clay (A) and sandy (B) soils.

Figure 6 Species richness (Chao1) and diversity (Shannon) of the fungal community in clay (A) and sandy (B) soils.

Multidimensional scaling analysis (MDS), using unweighted and weighted Unifrac distances (Hamady, Lozupone & Knight, 2010), were used to compare prokaryotic and fungal community composition in successive DNA extractions in both soil types. Detected prokaryotic community composition in clay soils showed to be similar in all three successive DNA extractions, regardless of the extraction kit used (Fig. 7A; Fig. S5A). When weighted Unifrac distance was used, clay 1 (PS) was the soil sample that presented the most different prokaryotic community among successive DNA extractions, with E1 and E3 being more similar to each other. However, when unweighted Unifrac distance was used, clay 3 (PS) showed to have the most different prokaryotic community among successive DNA extractions (Fig. S5A). Despite successive DNA extractions yielding similar prokaryotic community composition, some taxa clearly increased in relative abundance when multiple extractions were performed in a single sample. For instance, a clear increase in relative abundance of the taxa Thaumarchaeota and Firmicutes could be seen with successive DNA extractions in all clay soil samples (Table S3). Other taxa, such as Verrucomicrobia, Choroflexi and Acidobacteria, showed considerably higher relative abundance in E2 and/or E3 for some of the soil samples (Fig. 8A). Differently from prokaryotic communities, observed fungal communities in clay soils were different in successive DNA extractions (Fig. 9A; Fig. S6A). As observed for bacterial and archaeal communities, clay 1 showed the biggest variation in fungal community when comparing successive extractions. Successive extractions promoted an increase in relative abundance of a few fungal taxa in some of the clay soils (Fig. 10A), however, such increase showed to be kit dependent in most of the cases. Sandy soils, as clay soils, presented a similar prokaryotic community composition when successive extractions were compared (Fig. 7B; Fig. S5B). Of all three sandy soils, sandy 3 presented the biggest difference in prokaryotic community composition when comparing successive extractions. As observed for clay soils, some taxa were enriched when multiple extractions were performed in a single soil sample. Firmicutes and Planctomycetes increased in relative abundance with successive DNA extractions in all sandy soils (Fig. 8B; Table S3). Despite not having an increase in relative abundance in all soil samples, some taxa showed a considerable increase with successive DNA extraction, such as Actinobacteria, Chloroflexi and Proteobacteria (Fig. 8B). Fungal community composition in successive DNA extractions on sandy soils was rather similar, contrary to clay soils (Fig. 9B; Fig. S6B). However, for sandy 1 fungal community in E2 showed to be different than community present in E1 and E3, when analysis was performed using unweighted Unifrac distance (Fig. S6B). Increase in relative abundance of fungal taxa with successive extraction in sandy soils was also observed, with taxa Ascomycota presenting an increase in all sandy soil samples, regardless the DNA extraction kit used (Fig. 10B). A few bacterial and fungal taxa (phylum level) were observed only in E2 and/or E3, indicating that additional taxonomical groups can be identified with successive DNA extractions. However, the relative abundances of such taxa were always very low (lower than 0.05% for prokaryotes and 0.02% for fungi), and the occurrence of additional taxa was not observed for all soil samples (Table S4). Analysis of similarity (ANOSIM) was used to compare microbial communities in both soils, which had been obtained by different DNA extraction strategies. Prokaryotic community composition, in both soils, was not influenced by DNA extraction kit (p > 0.05). Fungal community composition, obtained with different DNA extraction kits, was significantly different only in clay soils (ANOSIM, p < 0.05, R = 0.210).

Figure 7 Multidimensional scaling (MDS) analysis of weighted Unifrac values from prokaryotic community in clay (A) and sandy soils (B).

Figure 8 Relative abundance of the five most abundant prokaryotic taxa found in clay (A) and sandy (B) soils.

Abundance is depicted as percentage, where one (1.0) corresponds to the sum of all taxa found.

Figure 9 Multidimensional scaling (MDS) analysis of weighted Unifrac values from fungal community in clay (A) and sandy soils (B).

Figure 10 Relative abundance of the five most abundant fungal taxa found in clay (A) and sandy (B) soils.

Abundance is depicted as percentage, where one (1.0) corresponds to the sum of all taxa found.

Discussion

Currently, characterization of complex microbial communities such as those present in soil relies heavily on the use of molecular approaches. Such approaches are often used to assess microbial abundance, community composition and diversity. Independently of the approach used, DNA contained in a soil sample needs to be separated from the soil phase as a first step. Therefore, soil DNA extraction is a crucial step, and the success of downstream processes used to characterize soil microbial communities will depend largely on how well this first step was performed. Previous studies have shown that not all microbial DNA contained in a soil sample is extracted with a single DNA extraction (Bürgmann et al., 2001; Feinstein, Sul & Blackwood, 2009; Jones et al., 2011). Here, we performed multiple successive DNA extractions on a number of representative soil samples to assess bias related to incomplete DNA extractions, using two widely used commercial soil DNA extraction kits. Moreover, different molecular techniques were applied to determine whether successive DNA extractions would lead to different apparent microbial communities.

DNA yield of all used soils was affected by successive DNA extractions. Only one out of twelve soils tested, clay 3 (PS), had more than 90% of all extracted DNA recovered after E1. All other soil samples showed recovery from 30% to 60%, of the total obtained DNA, in E1, which indicates that a significant portion of extractable soil DNA is left behind at the end of the first extraction. This supports results previously described in the literature (Feinstein, Sul & Blackwood, 2009; Jones et al., 2011). Previous work has shown that DNA yield decreased with successive DNA extraction when, after the first extraction, the soil pellet was washed with extraction buffer; which indicates that DNA obtained in the successive extractions would probably come from newly lysed cells (Feinstein, Sul & Blackwood, 2009). Soil samples were frozen between the E2 and E3, as extractions were continued the following day. Effects of freezing–thawing on cell lysis, which would consequently affect DNA yield, cannot be excluded. However, it is expected the size of this effect not to exceed that of the bead-beating procedure used prior and after freezing. Overall, DNA yield was significantly higher using FS compared to PS, around four times for clay soils and three times for sandy soils. This is in line with previous studies that have also shown that FS is more efficient than PS in extracting DNA from various soil types (Leite et al., 2014; Vishnivetskaya et al., 2014). Considering that the initial cell lysis step, when using PS, is much more extensive compared to FS (5.5 m s−1 for 10 min for PS and 6.0 m s−1 for 40 s for FS), this might be surprising at first sight, however, after cell lysis, released DNA will strongly interact with soil particles, which can influence DNA yield (Lombard et al., 2011). Romanowski, Lorenz & Wackernagel (1993) demonstrated that up to 80% of added DNA was found to be adsorbed to sediment in less than 20 min. DNA adsorption to soil particles may be increased by DNA shearing (Pietramellara et al., 2001); however, DNA fragmentation of both kits was very similar (Fig. S7). Therefore, reasons for a higher DNA yield when using FS may be that this kit promotes not only a better soil homogenization, improving disruption of soil aggregates, but also a better cell lysis and DNA desorption from soil components. Furthermore, FS may have a decreased DNA degradation compared to PS, once DNA is released from cells. Perhaps, a very extensive cell lysis step, as used for PS, is counterproductive, as it would allow for adsorption of DNA to soil particles for a longer period. Vishnivetskaya et al. (2014) showed that FS was more efficient than PS in genomic DNA recovery from a permafrost soil, which could also be attributed to higher bacterial cell lysis efficiency. Differences in bead to soil ratio has also been found to influence DNA yield while bead-beating is used for mechanical cell lysis (Bürgmann et al., 2001). However, DNA extraction kits did not significantly affect DNA yield when recovery percentage was taken into account, which shows that independently of the kit and bead-beating time, more soil DNA was consistently obtained with successive extractions.

Similarly to the DNA yield, cumulative bacterial and fungal abundances increased with successive extractions in all samples, confirming that microbial DNA present in a soil sample is not fully recovered with a single extraction. Similar results have been reported earlier, where for most of the soils analyzed bacterial and fungal abundances levelled off after three successive extractions (Feinstein, Sul & Blackwood, 2009). Jones et al. (2011) also found that bacterial abundance increased considerably when multiple DNA extractions were performed on the same soil sample. In the present study, however, there was a considerable discrepancy between results of DNA yield and observed abundances of bacteria and fungi. Although DNA extraction kit significantly influenced total DNA yield, it did not influence observed total bacterial and fungal abundances. Although not significant, bacterial and fungal abundances were always higher on clay soil samples extracted with FS, however, differences were much smaller than those observed for DNA yield, especially for clay 2. That may indicate that using FS more DNA of non-microbial origin could be extracted, or that not all microbial DNA extracted is amplified during qPCR, which could be due to specificity or purity issues. Bacterial abundance in a permafrost soil showed to be around four times higher in samples extracted with FS compared to PS (Vishnivetskaya et al., 2014). The opposite was observed for sandy soil, where bacterial and fungal abundances were always higher on samples extracted with PS, despite DNA yield being higher in samples extracted with FS. Lower microbial abundances in samples extracted with FS may be attributed to a lower DNA purity obtained when using this kit, which could inhibit enzymatic reactions (Wilson, 1997). Both 260/280 and 260/230 ratios, which indicate DNA purity, were consistently lower in sandy soil samples that were extracted with FS (Table S5). Differently from what was reported by Kuramae et al. (2012), pasture and arable soil samples presented higher microbial abundance compared to forest soils.

Fungal communities in successive DNA extractions, which were identified by T-RFLP analysis, varied greatly in clay soils, whereas in sandy soils, fungal communities were more similar to each other, especially for samples extracted with PS. Differences observed between soil types indicate that clay soils might promote a greater degree of protection to fungal cells, which could make cell lysis more difficult. These results differ from previous findings described on the literature, where fungal communities in successive DNA extractions of three different soil types (organic, clay and sand) were almost identical (Feinstein, Sul & Blackwood, 2009). Soils used by Feinstein, Sul & Blackwood (2009) had a similar sand and clay content to those used here, therefore, other soil properties may play a role. Furthermore, soils analyzed here may have a higher abundance of fungal spores, which could explain the differences observed among successive DNA extractions.

Ecological diversity measures, such as the Chao1 and Shannon indices, were used in order to determine whether apparent prokaryotic and fungal richness and diversity would change in successive DNA extractions. Chao1 is often used to estimate species richness (total number of species in a community), and it relies on the presence of singletons and doubletons, therefore, giving more weight to rare individuals (Hill et al., 2003). The Shannon index on the other hand takes into consideration not only the number of species in a community but also their relative abundance, also giving more weight to rare than common species (Hill et al., 2003). For both variables, soil type and extraction kit, increase of prokaryotic and/or fungal species richness with successive DNA extraction was observed. This suggests that more taxa/OTUs are obtained when successive DNA extractions are performed, therefore, presenting a more realistic picture of the microbial community in those samples. Those taxa/OTUs possibly represent also rare organisms, which can be present in low abundance in the soil for various reasons, including dormancy. It is known that dormancy is a common life history strategy in microbes (Jones & Lennon, 2010), and it might as well be that metabolic changes caused by such strategy, even in nonsporulating species, lead to changes in cell structure and morphology that make cells harder to lyse. Increase in diversity with successive DNA extractions was also observed, especially in fungal communities. Increase in Shannon index (diversity) indicated that with successive extractions, not only new taxa/OTUs were being observed, but also that taxa/OTUs observed in these successive extractions appeared at more similar relative abundances. Therefore, results from both diversity indexes suggest that extracting soil DNA only once does not give a realistic description of species richness and diversity.

To determine whether successive DNA extractions would yield different prokaryotic and fungal communities in clay and sandy soils, cluster analyses of the 16S and 18S rRNA gene sequencing data were performed. MDS plots revealed that prokaryotic communities obtained in E2 and E3 were similar to E1, for both soils. At phylum level, differences that were observed among successive DNA extractions, of the same soil sample, were in majority shifts in abundance, as observed before (Feinstein, Sul & Blackwood, 2009; Jones et al., 2011). Various phyla increased in relative abundance with successive DNA extractions, such as Thaumarchaeota and Firmicutes in all clay soils and Firmicutes and Planctomycetes in all sandy soils. Such increase in relative abundance may indicate that organisms belonging to these phyla are more difficult to lyse, which could be a consequence of their life strategy and/or morphological characteristics, such as sporulation. However, Feinstein, Sul & Blackwood (2009) found that Firmicutes, gram-positive bacteria that are well known for having the ability to form endospores, did not increase in relative abundance with successive DNA extractions in clay and sandy soils. It is important to mention that only the first and the sixth DNA extractions were sequenced in their study. As observed by Feinstein, Sul & Blackwood (2009), bacterial phyla that were not identified in E1, but identified in E2 and/or E3 were always in a very low relative abundance. Differently from prokaryotic community, cluster analyses of the 18S rRNA gene data retrieved from successive DNA extractions revealed that communities obtained in E1 were different from E2 and E3 only in clay soils. Such difference was seen when analyses were performed with both, unweighted and weighted Unifrac distances, which confirms that not all fungal taxa were recovered after a single extraction. The opposite has been reported in literature, where fungal communities analyzed by T-RFLP from six successive DNA extractions were very similar to each other (Feinstein, Sul & Blackwood, 2009). A low number of fungal taxa were identified only in E2 and/or E3, with the majority being unknown taxa, which may indicate that incomplete DNA extractions could prevent the identification of new fungal taxa. As for prokaryotic communities, major shifts in community composition were due to changes in relative abundance of fungal taxa. The use of both markers, 18S rRNA gene and ITS regions, showed similar results, however, the dissimilarity of fungal communities analyzed by T-RFLP was higher for a few soil samples. A possible reason for that is the higher variability of the ITS regions compared to the 18S rRNA gene, which allow for a better taxonomic differentiation (Anderson & Cairney, 2004).

In conclusion, successive rounds of DNA extraction from several representative soil samples, using two widely commercial DNA extraction kits, resulted in the identification of additional prokaryotic or fungal phyla in some of the analyzed soils. However, when identified, additional phyla were always present at very low relative abundance. Additionally, shifts in relative abundance of well-known groups of soil archaea, bacteria and fungi were observed. In some cases, changes in relative abundance were such that communities originating from the same soil sample, but from a different extraction, were seen as different communities, as indicated by MDS plots. Total bacterial and fungal abundance increased considerably with successive DNA extractions, confirming that not all soil DNA is extracted in a single extraction. Often, in microbiome studies, multiple parallel extractions of a sample are performed, and DNA originating from all extractions is pooled for further experiments. Such strategy attempts to reduce variability between extractions in order to provide a more realistic representation of the microbiome of that particular sample. However, such strategy would still fail to provide an accurate estimation of the relative abundance of microbial groups present in that particular microbiome. Therefore, as Feinstein, Sul & Blackwood (2009), we argue that in order to improve microbial characterization, leading to a more comprehensive microbiome analysis, DNA should be obtained from multiple successive extractions of the same soil sample. DNA obtained from multiple successive extraction should be pooled previously being used in further experiments. Such approach may reduce DNA extraction bias and provide a better overall picture of the soil microbial community under analysis.

Supplemental Information

Supplemental Information 1 Supplementary Information.

Click here for additional data file.

We thank Henk Martens and Lucas William Mendes for the assistance with the T-RFLP experiments and analysis. Publication number 6220 from the Netherlands Institute of Ecology (NIOO-KNAW).

Additional Information and Declarations

Competing Interests

Author Contributions

Data Deposition

Hauke Smidt and Johannes A. van Veen are Academic Editors for PeerJ.

Mauricio R. Dimitrov conceived and designed the experiments, performed the experiments, analyzed the data, contributed reagents/materials/analysis tools, wrote the paper, prepared figures and/or tables, reviewed drafts of the paper.

Annelies J. Veraart analyzed the data, wrote the paper, prepared figures and/or tables, reviewed drafts of the paper.

Mattias de Hollander analyzed the data, contributed reagents/materials/analysis tools, wrote the paper, reviewed drafts of the paper.

Hauke Smidt contributed reagents/materials/analysis tools, wrote the paper, reviewed drafts of the paper.

Johannes A. van Veen conceived and designed the experiments, wrote the paper, reviewed drafts of the paper.

Eiko E. Kuramae conceived and designed the experiments, contributed reagents/materials/analysis tools, wrote the paper, reviewed drafts of the paper.

The following information was supplied regarding data availability:

Raw sequencing data have been submitted to the NCBI sequence read archive (SRA) under the accession numbers SRR5040745 and SRR5043664.

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
