# Peer review of "Successive DNA extractions improve characterization of soil microbial communities"

_PeerJ, doi:10.7717/peerj.2915_

## Round 0.1 · original submission · Major Revisions

Dear authors

Although some of the reviewers indicated an important lack of the novelty in your study, PeerJ Decisions are not made based on any subjective determination of impact, degree of advance, novelty, or for being of interest to only a niche audience. So, I cannot consider this as a lack of merit for your work. However, apart from that some reviewers encountered a substantial amount of details in the materials and methods and result sections that need to be amended. Specially, some of the results are considered not being adequately validated by the provided data and/or analysis and suggestions to improve them are given. I encourage you to take into account all comments driven by the reviewers and try to answer them in order to improve your manuscript.

Reviewer 1 ·

Basic reporting

The article shows the effect that successive DNA extractions, applied on a single soil sample, has on the quantity and diversity data currently used for the characterization of microbial communities. Even though the manuscript is well organized, with background, objectives, results and discussion coherent, the results presented are not novel.

Experimental design

Even though the analysis was conducted in order to compare two extraction methods applied by two noted commercial kits on soil chosen from the most important land management practices in the Netherlands (conventional and organic arable field, pasture, pine and deciduous forest), previous studies have focused on the comparative analysis for the same kits on similar types of soils. The effect that successive DNA extraction have on the yield and diversity was also previously reported.

Validity of the findings

Even if the data produced are numerous, robust and obtained by traditional and modern analytic approaches currently used for biodiversity studies, the results (at exceptions for some individual reports) do not contribute to novel findings than those previously reported. At exception of additional incubations times between extraction steps, none news on the extraction procedures is presented.

Additional comments

The manuscript could improve if data are presented in order to describe, in a more detailed form, the effects that DNA extraction soil methodology has on the biodiversity of the most important land management practices.

Reviewer 2 ·

Basic reporting

No Comments

Experimental design

No Comments

Validity of the findings

No Comments

Additional comments

In this paper from PeerJ Journal (Successive DNA extractions improve characterization of soil microbial communities by Dimitrov and colleagues) the authors propose an interesting study and valuable to the field of soil microbiology: soil DNA extraction as a critical step to characterize the soil microbial communities. The topic of the paper fits the scope of the journal. I think that some of the questions identified in the introduction are very interesting and have not been well studied by the scientific community. However, there are several limitations of the approach that limit the author's ability to address those interesting questions. Mainly, I have serious concern about the experimental design. The results of DNA yield and qPCR sounds good since a correct experimental design with adequate number of replications and statistical treatment has been carried out, although in my opinion the sampling process would be improved with more replicates at the same location; sampling processes may influence alfa and beta diversity estimates. However, microbial community analysis (t-RFLP and pyrosequencing ) are based in a single replicate. Technical replicates are needed to statistically correct intra-sample variation, and field-base replicate samples are desirable to substantiate results. I fell that the authors should demonstrate how it is possible to compare their samples without, for example, a beta diversity analysis.
1.- Which modifications, and why? line 104
2.- 10 ng DNA / 10 ul for qPCR (lines 139-142) , half DNA concentration ( 25 ng DNA / 50 ul) for pyroseq (lines 176-179). Why?
3.- Give the appropriate reference for CANOCO (ter Braak and Smilauer 2002) (line 207)
4.- Taxonomic (line 221)
5.- Cancel (average…) line 251
6.- Lines 258, 260, 261. Figure 2C instead of 1C.
7.- Try to explain the different behaviour of clay 3, (higher OM content?), line 291
8.- clay 2, line 353
9.- Justify “data not shown”, line 465
10-Lines 491-493, too speculative
Figures 2, 3 and 4 need to show the statistical significant

Reviewer 3 ·

Basic reporting

The manuscript describes the effect of DNA extraction from different soils in three successive steps on depiction of bacterial and fungal communities. To characterize microbial communities various molecular techniques based on 16S/18S rRNA genes were applied including determination of copy numbers by quantitative real-time PCR and community profiling by T-RFLP and 454 amplicon sequencing.

First of all, both approaches comparing commercial soil DNA extraction kits as well as subjecting different soils to successive DNA extraction steps has been reported previously. The majority of those studies employed similar methods to characterize microbial communities as described here. The present report and the paper of Feinstein et al. 2009 strongly resemble each other. Moreover, the authors couldn’t satisfyingly prove that the described approach provides significant benefits for studies to characterize microbial communities in soils. Thus, the scientific community would not greatly profit from the reported experimental protocol.

Experimental design

DNA extraction: The intermediate freezing step between the 2nd and 3rd extraction step could potentially have an additional effect on the extraction efficiency, but is completely ignored by the authors. Does it matter whether the samples are frozen in between or not?
DNA yield: Two methods (Qubit 2.0 fluorometer; NanoDrop 1000 spectrophotometer) to quantify DNA yields were mentioned in Material&Methods. Usually the results differ and it should be stated which data are indicated in Figure 2 and Table S2?

Validity of the findings

Two of the main conclusions stated by the authors, both based on amplicon sequencing, are not adequately validated by the provided data and/or analysis.

(I). Successive DNA extraction “can lead to a better picture of overall community composition”: How? Here, the DNA eluates from three extraction steps are analyzed as single entities, and, indeed, the relative abundances of certain phyla change. However, the direction and degree of change from sample to sample and from extraction step 1-2 and 2-3 doesn’t appear to follow any inherent role. How robust are those changes? The relative abundance numbers lacks indication of deviations (Fig. 8, Table S3). To calculate correlation indices, suitable statistical tools should be applied. To create a better and realistic picture of the microbiome, which of the extraction step reflect the reality? Should be the taxonomic compositions of all three extractions merged? These questions should be answered and supported by mathematical means. However, to study the relative distribution of the main taxa, would it be more meaningful to combine the DNA extracts from all three steps prior to amplification for sequencing?

II) “successive DNA extractions did reveal several additional microbial groups”: The authors consciously skipped the step of normalizing read counts. But, the presence or absence of certain, particularly low abundant taxa ultimately depends on the sequencing depth. Indicating taxa uniquely present in E2 and E3 is only acceptable after normalization. Similar, the comparison of alpha diversity metrics (Fig. 5 &6) is only valid based on normalized OTU tables.

I strongly recommend a binomial OTU-based analysis. Compared to the relative abundance-based analysis, an approach only based on presence/absence information of certain microbial groups is probably more reliable to prove the methodological significance of successive DNA extraction for microbiome research. An OTU-based analyses would be required e.g. enabling the identification of unique/shared or additional OTUs. The calculation of a rarefraction curve could lead to an indication how successive DNA extraction affect the coverage of the expected number of OTUs.

---

## Round 0.2 · accepted · Accept

Dear authors,

Thank you for taking into account all comments made by the reviewers and providing with more supporting information and many clarifications. I think your manuscript has improved and is now ready to be published.

Reviewer 2 ·

Basic reporting

Dear Editor,

My concerns from my previous review have been addressed.

Experimental design

ok

Validity of the findings

ok